# Astronomy as a Ground-Truth Sandbox for Interpreting Large Models

## Abstract

Studies of mechanistic interpretability often target concepts like honesty or deception, where ground truth is ambiguous. We propose a comprehensive (Nanda et al., 2023) testbed: astronomical observations with physically-defined labels. Using 74 925 galaxy images with spectroscopic redshifts that measure astronomical distance, we study how this quantity is represented across three architectures (DINOv2, Qwen2-VL, AstroPT). We test four common assumptions and find: (i) distance is linearly decodable yet not axis-aligned, instead concentrated in a low-rank subspace; (ii) cross-model geometry can align while usable linear features do not transfer (a geometric-functional paradox); (iii) steering along learned distance directions causally shifts distance-related language, with strong prompt dependence; and (iv) breaking a monitor via linear removal is not a certificate of deletion under adversarial audit. Overall, astronomy provides a grounded sandbox that makes interpretability claims falsifiable under controlled manipulations.

## 1 Mechanistic interpretability, fuzzy concepts, and astronomy

Mechanistic interpretability has matured rapidly as a field, yet it remains haunted by a fundamental problem: the concepts we most want to understand (such as honesty, deception, and intent) lack an unambiguous ground truth (Lipton, 2016; Doshi-Velez & Kim, 2017). In other words, when a probe claims to detect 'truthfulness' we cannot easily rule out that it has learned a correlated proxy like confidence or verbosity. We propose using astronomical data as a complementary test-bed that makes these claims experimentally falsifiable. This is feasible as astronomical observations bring with them target variables that are physically defined and independently measured. Here, we explore a representative case study of the above idea: we focus on galaxy redshift ($z$), a continuous, precisely measured proxy for distance, and use it as a controlled sandbox for testing interpretability assumptions under interventions and known confounders. Concretely, we ask four questions: whether a concept behaves like a single direction or a higher-rank subspace (tested via 1D-vs-full linear probes and a $k$-sweep); whether cross-model geometric alignment implies usable feature transfer (tested via alignment, transfer, and subspace comparisons); whether diagnostic directions are also causal levers on behavior (tested via steering with matched controls) (Turner et al., 2023); and whether removing a direction or subspace constitutes deletion (tested by monitor-breaking versus post-edit adversarial re-learning and guarded tradeoffs). Our goal is to make interpretability claims more testable by working in a setting with objective labels and known confounders.

## 2 Dataset, representations, and models

We use the `Smith42/galaxies` dataset hosted on Hugging Face[1] which consists of the galaxy images from DESI Legacy Survey DR8 survey (Dey et al., 2019). It contains 8.7 million galaxy cutout images, paired with metadata such as redshift, photometry, and morphology. In the dataset release, we use the cropped cutouts ($256 \times 256$ pixels), which corresponds to a field of view of roughly 1.1 arcmin at the standard Legacy Surveys pixel scale. We use $N = 74,925$ galaxies from Legacy Survey DR8 with paired redshifts from DESI, restricted to $0.1 \leq z \leq 0.5$ to ensure reasonably resolved morphology. We use a separate validation set consisting of 86 500 galaxies.

---

[1]https://huggingface.co/datasets/Smith42/galaxies

We are not trying to build the best redshift predictor. We use redshift as a clean target to test whether common interpretability techniques (readout directions, subspace alignment, steering, erasure, and monitoring) (Alain & Bengio, 2016; Raghu et al., 2017) behave the way we expect them to do. To show that our techniques are general, we test models that are different in training signals, objectives, and inductive biases (Huh et al., 2024). If a distance concept is really tied to the world, we might expect some overlap across these very different models. If, instead, each model builds its own internal basis for the same target, we should see cases where global geometry lines up but feature transfer fails. A major failure mode in mechanistic studies is silently mixing representation locations (final layer vs. peak layer vs. interface outputs), which can make results look internally inconsistent (Alain & Bengio, 2016; Belinkov, 2021). For each model, we extract our visual embeddings from the recommended layer in the original model's documentation.

With all the above in mind we test a vision-only foundation model, a vision-language model, and an astronomy specialist: **(1) DINOv2** is a ViT-based vision encoder trained with self-supervision to produce broadly transferable visual features. We use it as a vision-only baseline: if redshift is present here, it must be mediated through visual factors (photometry and morphology, plus any pipeline-induced cues such as effective resolution/sharpness from the rescaling of galaxy cutout images), rather than language-conditioned shortcuts (Oquab et al., 2023). **(2) Qwen2-VL** is a VLM with a vision tower feeding a language model through an interface/merger. This lets us ask not only whether distance is decodable, but where it becomes usable inside the vision tower and whether it causally influences language behavior under interventions such as steering or patching Wang et al. (2024). **(3) AstroPT** is a domain-specialized model trained for astronomical imagery. We use it as a specialist reference point: it helps separate effects that come from generic web-vision pretraining from effects that persist in models fine-tuned to astrophysical data distributions (Smith et al., 2024).

Once extracted, we standardize each dimension through z-scoring (for feature $j$, $x'_j = (x_j - \mu_j)/\sigma_j$, where $(\mu_j, \sigma_j)$ are computed on the training set and then held fixed for validation/test[2]). Our primary metric is test-set $R^2$ of a ridge probe (linear regression with $L_2$ regularization) (Sasaki et al., 2025) predicting $z$ from embeddings.

We use two families of distance directions. First, we used difference-in-means direction ($v_z = \mu_{\text{high } z} - \mu_{\text{low } z}$ using top/bottom quartiles). Secondly, we used partial least squares (PLS) (Assunção & Afonso Fernandes, 2024) physics subspace (PLS on standardized embeddings with varying number of components $k$). We use these two choices because they answer two different questions about how a concept is represented. The difference-in-means vector $v_z = \mu_{\text{high } z} - \mu_{\text{low } z}$ is the simplest diagnostic test that asks whether distance looks like a single dominant linear direction. It is closely related to classic discriminant-style directions (Dorfer et al., 2015; Radford et al., 2015) that separate two groups by their mean shift, and it is easy to interpret and use for steering because it does not require fitting a complex model.

## 3 RESULTS AND CONCLUSIONS

**Linear decodability.** We first test whether redshift behaves like a single linear feature (such as the board state directions in Othello-GPT (Li et al., 2022; Nanda, 2023)). For each model's final visual embedding, we compare: (i) a full Ridge probe on the entire embedding ($R^2_{\text{full}}$); (ii) a 1-D probe on the projection onto $v_z$ ($R^2_{\text{1D}}$); and (iii) a random baseline (performance when regressing on a random unit vector projection).

| Model | $R^2_{\text{full}}$ | $R^2_{\text{1D}}$ | Random $R^2$ | $\Delta R^2$ |
|---|---|---|---|---|
| AstroPT (specialist) | 0.194 | 0.010 | 0.004 | 0.184 |
| DINOv2 (vision) | 0.291 | 0.066 | 0.003 | 0.225 |
| Qwen2-VL (VLM) | 0.498 | 0.108 | 0.001 | 0.390 |

Table 1: Linearity gap for redshift. Larger gaps indicate that distance is less well approximated by a single direction.

Table 1 shows that language supervision seems to help: Qwen2-VL has far larger linearity in its embedding space compared to DINOv2. Our results show that physical concepts often reside in a

---

[2] https://scikitlearn.org/0.20/auto_examples/preprocessing/plot_all_scaling.html

higher-rank manifold rather than a single direction (linearity gap), consistent with hypothesis that concepts occupy subspaces rather than single directions (Elhage et al., 2022; Park et al., 2023). Next we will probe the size of each model's physical subspace.

We measure how our redshift embedding linearity changes as we increase the dimension $k$ of a PLS subspace. For $k \in \{1, 2, 4, 8, 16, 32\}$ we fit PLS on training embeddings and train a linear probe in the resulting subspace. Figure 1 shows that Qwen2-VL captures most of the linearly decodable redshift information in the first $\sim$16 dimensions, suggesting a compact physics subspace rather than a single direction. Basic image statistics explain only part of this subspace: regressing magnitude (a logarithmic measure of how bright a galaxy looks in a given band (lower magnitude = brighter object)) onto the 16-D subspace yields $R^2 \approx 0.13$ (vs $\approx 0.01$ for a random subspace), and color carries a smaller but non-zero signal ($R^2 \approx 0.02$). This large residual gap implies that the model does not rely solely on brighter equals closer heuristics. Instead, the subspace must encode higher-order visual cues—such as angular size, morphology, which act as learned proxies for distance. We further show that the best linear redshift monitors are often not at the final interface layer, and that Qwen2-VL exhibits an interface-layer collapse that can make output probes structurally unseen as shown in Appendix B; Fig. 4–5.

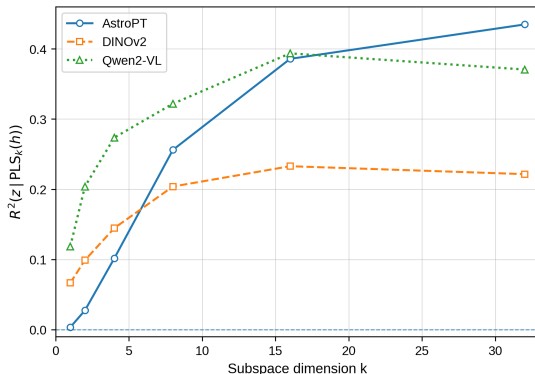

Figure 1: PLS tomography: test $R^2$ vs number of components $k$. Qwen2-VL saturates quickly (around $k \approx 16$ with $R^2 \approx 0.38$). DINOv2 plateaus around $R^2 \approx 0.23$. AstroPT continues improving to $R^2 \approx 0.44$ at $k = 32$.

**Cross-model transfer and Steering.** Based by the previous results, we can now ask: even if two models both encode distance, do they use the same internal feature basis for it? To understand this we first compute a redshift direction for each model and align embeddings to AstroPT via an orthogonal Procrustes transform (Ling, 2021) on the galaxies. Next,

Table 2: Cross-model physics alignment: moderate geometric similarity but poor functional transfer. $\Delta_{\text{cosine}}$ is the cosine distance,

| Model pair | $\Delta_{\text{cosine}}$ | Transfer $R^2$ |
|---|---|---|
| DINOv2 $\rightarrow$ AstroPT | 0.6587 | 0.0167 |
| Qwen2-VL $\rightarrow$ AstroPT | 0.6239 | 0.0440 |

we measure geometric similarity (cosine similarity of aligned directions) and functional transfer ($R^2$ when probing one model using AstroPT's direction). Finally, we compare subspaces using principal angles. The pattern seen in Table 2 is consistent with the Platonic Representation Hypothesis (Huh et al., 2024): geometry suggests a shared manifold, yet feature transfer fails because of low transfer $R^2$ (found using AstroPT's direction to probe the other model's aligned embeddings). This is the 'geometric-functional paradox': spaces can be globally alignable while the specific linear features that implement a concept remain model-specific (Kornblith et al., 2019; Raghu et al., 2017).

The above shows a compact redshift-predictive subspace and a concrete direction $v_z$ that reads out part of it; we now test whether this signal is causally usable for behavior in a VLM. We steer Qwen2-VL by adding a scaled shift along the learned direction at the vision$\rightarrow$language interface: $h' = h + \alpha \sigma v_z$, where $v_z$ is the unit-normalized redshift direction and $\sigma$ is the empirical standard deviation of $\langle h, v_z \rangle$ over the dataset (so $\alpha$ is dimensionless). We sweep $\alpha \in [-5, 5]$ and measure (i) the mean redshift predicted by a fixed linear probe and (ii) a caption redshiftiness score defined by a forced-choice prompt, as the signed log-odds $\log p(\texttt{distant}) - \log p(\texttt{nearby})$ (reported after a bounded normalization). Figure 2 shows a sharp, monotonic shift under $v_z$ but not under a matched-norm random direction, indicating that the learned distance direction is not merely diagnostic.

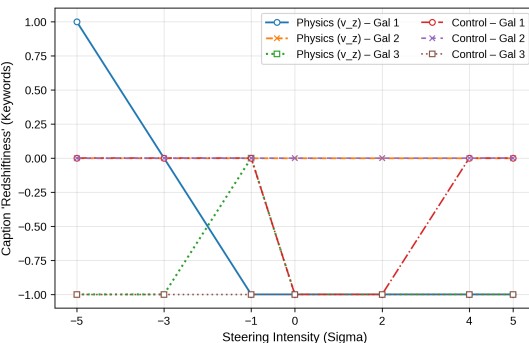

Figure 2: Interface steering in Qwen2-VL. Redshiftiness is a forced-choice caption score based on signed log-odds between distant vs. nearby. Steering along $v_z$ shifts distance-related language monotonically; a matched-norm random control does not.

**Erasure.** So far we have treated redshift as a single concept: does the embedding contain information about distance, and can a monitor read it out? Now we ask, if a model predicts redshift well, is it using (i) real photometric cues that are physically tied to distance (brightness, color), or (ii) a non-physical shortcut channel that happens to correlate with redshift in the dataset? Additional evidence that simple linear subspace removal can 'break the monitor' while leaving recoverable redshift signal appears in Appendix A (Fig. 3, Table 4). We also quantify redshift-photometry overlap and the resulting tradeoffs for guarded removal are shown in Appendix D (Tables 5-8, Appendix D.1).We approach this question this by splitting the redshift signal into two parts: **(1) A physics-entangled component:** the part of the redshift signal that overlaps strongly with real photometry (magnitude, color). Editing it should inevitably harm photometry. **(2) a 'rote' component:** the part of the signal that is separable from photometry and responds to background/context statistics (e.g., crowding, clutter, field complexity). Editing it should reduce redshift predictability while largely preserving photometry.

| Model | Baseline $R^2(z)$ | Final $R^2(z)$ | Bottleneck |
|---|---|---|---|
| Qwen2-VL | 0.414 | 0.200 | color damage triggers |
| DINOv2 | 0.242 | 0.242 | guard triggers immediately |
| AstroPT | 0.479 | 0.382 | color damage triggers |

Table 3: Guarded erasure across models (protecting photometry).

We develop a novel method: Iterative Orthogonal Refinement (IOR) and compare with a standard baseline method called LEACE (Belrose et al., 2023)(perfect linear removal). IOR is a greedy, guarded procedure. At each step we fit a linear probe to predict redshift from the current embedding. We treat the probe weight vector as the current most predictive redshift direction and project it out. Crucially, we stop early if protected variables (real photometry) degrade beyond a tolerance. This is conceptually similar to iterative nullspace projection methods (Esmaeili et al., 2016; Ravfogel et al., 2020), but with an explicit protected task constraint. LEACE provides a closed-form linear transformation that removes all linearly decodable information about a concept (here redshift) while minimizing representational change. We use it as a reference point: what perfect linear removal looks like when we do not enforce protected-task constraints, and why perfect $z$-removal may inevitably damage photometric variables when concepts overlap. We repeat the guarded erasure analysis across models as shown in Table 3. Qwen2-VL shows the largest removable component under the photometry constraint. AstroPT shows a smaller removable component. DINOv2 shows essentially no removable component under the same guard: attempting to remove redshift immediately harms photometry, consistent with stronger entanglement (more details in Appendix D, E).

**Conclusions.** Astronomical redshift gives a physically grounded, continuously labeled target with measured confounders, letting us falsify common interpretability assumptions under controlled interventions. We find evidence that redshift is linearly decodable but concentrated in a low-rank subspace, that cross-model alignment can preserve geometry while linear features fail to transfer: a geometric-functional paradox, that interface steering causally shifts distance language, and that monitor-breaking via linear erasure does not certify deletion, since redshift can remain recoverable and monitoring is layer-dependent. These are the same failure modes we worry about for 'fuzzy' LLM concepts (such as honesty or deception), but astronomy makes them testable by turning selection effects, photometry, and context into explicit confounders and intervention targets.

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

## APPENDIX ROADMAP

Appendix A audits linear erasure (monitor-breaking vs. residual knowledge), Appendix B studies layerwise monitoring and interface collapse, Appendix C tests OOD robustness under a morphology shift, and Appendix D analyzes photometry–redshift entanglement and guarded erasure details.

## A   LINEAR CONCEPT ERASURE, AND WHY IT FAILS UNDER AUDIT

We attempt to erase distance by projecting out an 8-D physics subspace $U$:

$$h_{\text{clean}} = h - U(U^\top h).$$

This breaks the original distance monitor dramatically: the baseline probe drops from $R^2 \approx 0.41$ to $R^2 \approx -4.5$, while erasing a random 8-D subspace yields only a modest drop to $R^2 \approx 0.19$.

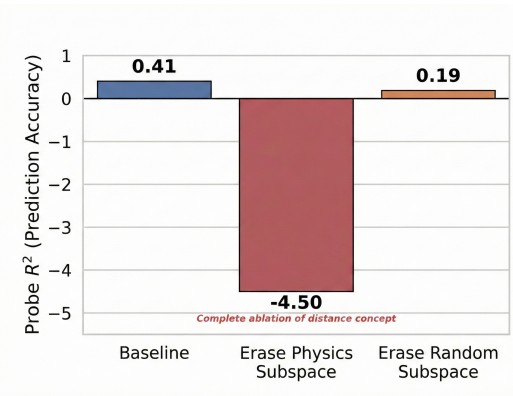

Figure 3: Linear subspace erasure. Removing the learned physics subspace breaks a fixed redshift monitor far more than removing a random subspace of equal dimension.

However, we then run an adversarial audit by treating $h_{\text{clean}}$ as a clean representation and train new probes post-erasure. A linear attacker still achieves $R^2 \approx 0.36$, and a small MLP attacker achieves $R^2 \approx 0.23$ (control MLP on random-erased embeddings reaches $R^2 \approx 0.30$).

Table 4: Adversarial auditing of concept erasure (Qwen2-VL). Breaking a monitor is not the same as deleting the concept.

| Attack scenario | Probe type | $R^2$ |
|---|---|---|
| Physics-erased embeddings | Ridge (linear) | 0.36 |
| Physics-erased embeddings | 2-layer MLP | 0.23 |
| Random-erased embeddings | 2-layer MLP | 0.30 |

From table 4 we observe that a monitor can fail while the model still knows distance in a form recoverable by another monitor. This mirrors a core ELK concern: monitor failure is a weak guarantee about latent knowledge.

## B   WHERE IN THE NETWORK SHOULD WE MONITOR?

### B.1   TRAJECTORY OF EMERGENCE: PHYSICS PEAKS IN THE MIDDLE

For Qwen2-VL and DINOv2, we probe multiple visual blocks and report $R^2$ vs depth (normalized so 0 is input, 1 is final visual layer). Qwen2-VL peaks around layer 24 ($R^2 \approx 0.40$) and then collapses at the final visual layer ($R^2 \approx 0.04$). DINOv2 peaks near the penultimate block ($R^2 \approx 0.20$) and becomes slightly negative at the final layer.

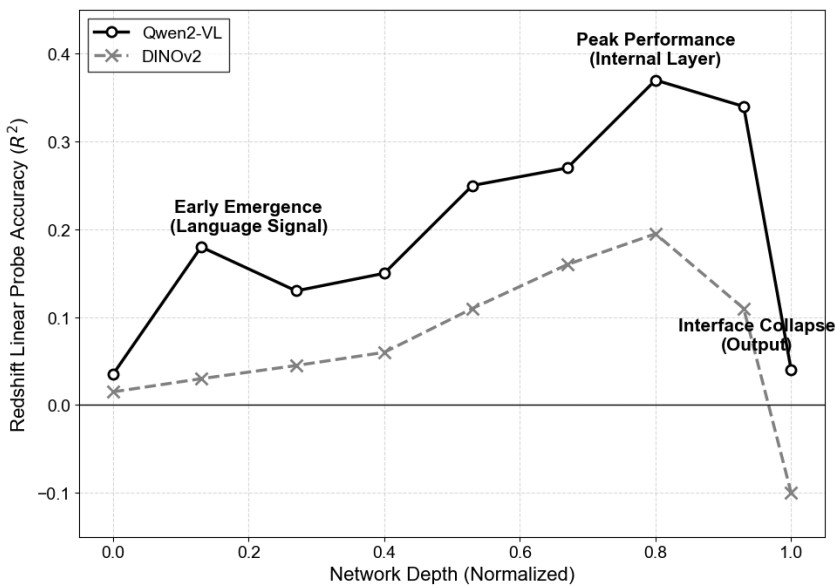

Figure 4: Layerwise redshift decodability. Best monitors are often not at the interface layer.

## B.2 INTERFACE COLLAPSE: RANK BOTTLENECKS CAN SCRAMBLE LINEAR STRUCTURE

To understand the collapse, we compare Qwen2-VL layer 24 (peak) to layer 31 (final visual output). Two hypotheses are plausible: (i) dilution (signal drowned by variance) or (ii) dimensional collapse (representation becomes low-rank).

Empirically, normalized signal strength along the redshift direction does not decrease at the final layer, while the representation becomes effectively low-rank: a small number of principal components explain most variance, and a PCA manifold view shows clumping and loss of a smooth redshift gradient.

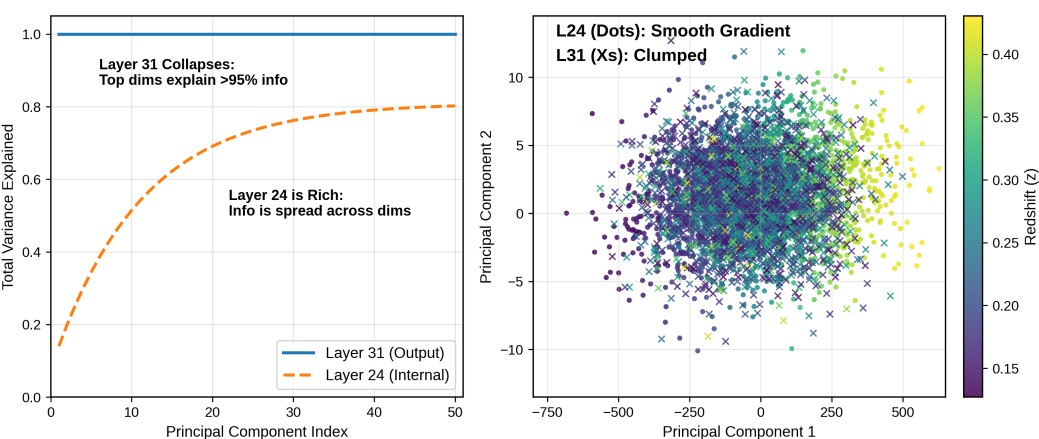

Figure 5: Dimensional bottleneck and manifold collapse. Left: cumulative variance explained at layer 24 vs layer 31. Right: 2-D PCA colored by redshift shows a smooth gradient at layer 24 but clumped, scrambled structure at layer 31.

From a monitoring point of view, this suggests the presence of a blind spot since the output layers probes can be structurally blind to internal concepts when interface layers compress and remap representations.

## C   Do linear monitors generalize out of distribution (OOD)?

We evaluate OOD robustness via a morphology shift (Galaxy Zoo-style): train on smooth (elliptical galaxides) and test on featured (spiral galaxies), and vice versa. Training on smooth yields $R^2 \approx 0.59$ in-domain and $R^2 \approx 0.30$ OOD. Training on spirals yields $R^2 \approx 0.86$ in-domain but collapses to $R^2 \approx 0.11$ OOD.

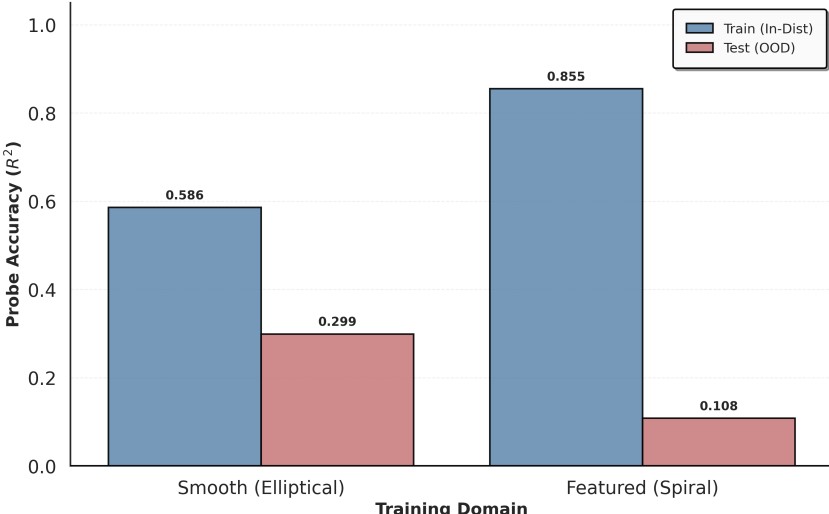

Figure 6: OOD robustness under a morphology shift. High in-distribution $R^2$ can be misleading: the spiral-trained monitor fails on smooth galaxies.

This gives us an important lesson for safety monitoring: style shifts can break monitors that look excellent in-distribution.

## D   Subspace entanglement in Qwen2-VL: redshift overlaps heavily with photometry

Before erasing anything, we measure how much the redshift subspace overlaps with brightness and color subspaces using real labels. We compute overlap scores by comparing the top-$k$ subspaces induced by linear probes (details in Appendix).

Table 5: Subspace overlap in Qwen2-VL (higher means more shared linear geometry).

| Concept pair | Overlap | Interpretation |
|---|---|---|
| Redshift–Brightness | 0.726 | strongly entangled |
| Redshift–Color | 0.820 | very strongly entangled |

Table 5 shows us that in Qwen2-VL, distance is not isolated. A large fraction of the redshift-predictive geometry is shared with photometry. This makes clean unlearning fundamentally hard: removing $z$ will often remove parts of magnitude and color.

### D.1   Selective erasure in Qwen2-VL: how much redshift can we remove without breaking photometry?

We apply IOR to Qwen2-VL with photometry as protected labels. In the current run, IOR removes a substantial portion of the redshift signal before it starts damaging photometry.

Table 6: IOR on Qwen2-VL with photometry guard.

| Metric ($R^2$) | Baseline | After IOR | Change |
|---|---|---|---|
| $z$ | 0.414 | 0.200 | 50.1% removed |
| mag | 0.287 | 0.276 | small drop |
| color | 0.343 | 0.292 | moderate drop (hits guard) |
| # directions removed | 0 | 5 | — |

A conservative interpretation is that Qwen2-VL contains a redshift component that can be removed with only mild photometry damage. We refer to the first IOR direction as the rote vector and define:

$$\texttt{rote\_score}(x) = \langle h(x), v_{\text{rote}} \rangle,$$

where $h(x)$ is the embedding and $v_{\text{rote}}$ is the first removed direction (unit-normalized).

**Adversarial audit:** After IOR, we retrain probes on the edited embeddings. A linear probe still achieves $R^2 \approx 0.20$ by construction, and a small MLP also recovers additional signal. This is consistent with a mixed picture: we can remove part of the separable linear shortcut component, but non-linear (and photometry-entangled) information remains.

Table 7: Auditing after IOR on Qwen2-VL.

| Probe | $R^2(z)$ (baseline) | $R^2(z)$ (after IOR) |
|---|---|---|
| Ridge (linear) | 0.498 | 0.200 |
| MLP (nonlinear) | — | 0.14 |

## D.2 LEACE VS. IOR: PERFECT $z$-REMOVAL DAMAGES PHYSICS

We compare IOR to LEACE($z$). LEACE achieves near-zero linear $z$ information, but it also harms photometry, which is expected given the large overlap in Table 5.

Table 8: Trade-off between erasure and preservation in Qwen2-VL.

| Method | $R^2(z)$ | $R^2(\text{mag})$ | $R^2(\text{color})$ |
|---|---|---|---|
| Baseline | 0.498 | 0.290 | 0.349 |
| IOR (guarded) | 0.178 | 0.224 | $\approx 0.00$ |
| LEACE($z$) | $\approx 0.00$ | 0.174 | 0.056 |

These results have two main implications. Firstly, LEACE shows that linear $z$-removal is possible, but it is not free: when concepts share geometry, removal deletes shared structure. IOR removes less $z$, but the color metric is sensitive; this is a sign that the guard and evaluation need to be aligned if we want to claim clean separation.

## D.3 WHAT DOES THE ROTE VECTOR CAPTURE? (IMAGES, PROXY CORRELATIONS, AND INTERVENTIONS)

We next try to make the rote vector interpretable. Figure 7 shows images with high vs. low activation along the top rote direction. For Qwen2-VL and AstroPT, high-activation examples tend to look crowded or cluttered (many small sources / complex background), while low-activation examples look cleaner and sparser. For DINOv2 we do not find a separable rote direction under the same photometry constraint, so we do not show a "DINO rote" panel.

**Proxy correlations:** We compute simple image statistics and compare them to rote_score: number of bright peaks (n_peaks, a crowding proxy), pixel standard deviation (contrast/noise

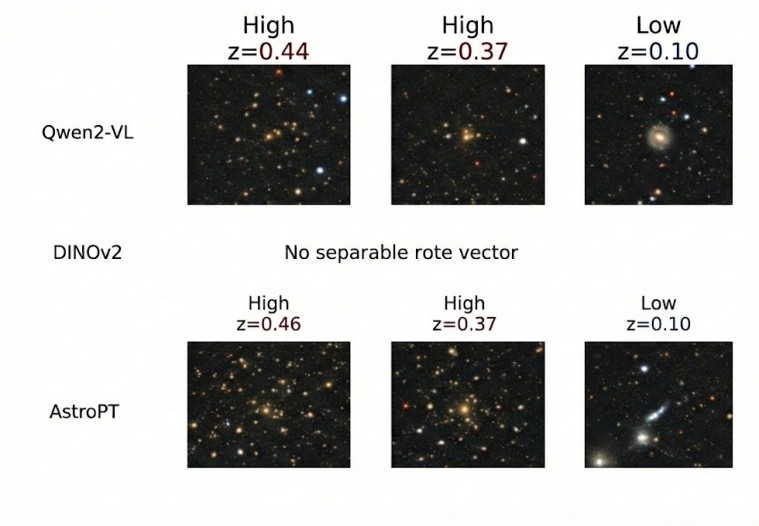

Figure 7: Examples of images that score high (red) vs low (blue) on the most erasable redshift-related direction (the first IOR/rote vector) for three models. For Qwen2-VL and AstroPT, high-activation images tend to look more crowded/complex (many small sources or clutter), while low-activation images look cleaner/sparser—suggesting this direction may capture a shortcut cue (scene crowding) that correlates with redshift. DINOv2 shows no separable rote vector under the same constraint (attempting to remove redshift ($z$) immediately harms photometry), so no clear shortcut direction is displayed.

proxy), $p99$ (bright-tail proxy), and mean background. The clearest relationship is between `rote_score` and `n_peaks`: higher rote activation is associated with denser fields. The relationships with mean/$p99$/std are weaker, which suggests that the rote vector is not simply "brighter image" but something closer to crowding or field complexity.

**Causal validation by intervention:** Correlations can be misleading, so we test whether the rote feature responds to controlled perturbations. We re-embed images in Qwen2-VL after two interventions: (i) synthetic clutter injection (adding faint sources) and (ii) center masking (replace most of the image outside the center with background). Both interventions systematically increase `rote_score`, and they also increase the probe-predicted redshift. In this causal test, when we modify background/context statistics, the rote feature moves, and the model's redshift readout moves with it.

### D.4 CROSS-MODEL SEPARABILITY: QWEN2-VL VS. ASTROPT VS. DINOV2

We repeat the guarded erasure analysis across models. Qwen2-VL shows the largest removable component under the photometry constraint. AstroPT shows a smaller removable component. DINOv2 shows essentially no removable component under the same guard: attempting to remove redshift immediately harms photometry, consistent with stronger entanglement.

## E VISION-ONLY ENTANGLEMENT IN DINOV2

Qwen2-VL shows a separable shortcut direction that looks like crowding/clutter. DINOv2 behaves differently. Because DINOv2 is trained with self-supervision and no language supervision, it is plausible that it encodes redshift only through broad visual factors that are already physically correlated with distance (brightness, color, apparent size/morphology). In that case, there may be no clean "rote" component to remove without harming the same representation that encodes photometry.

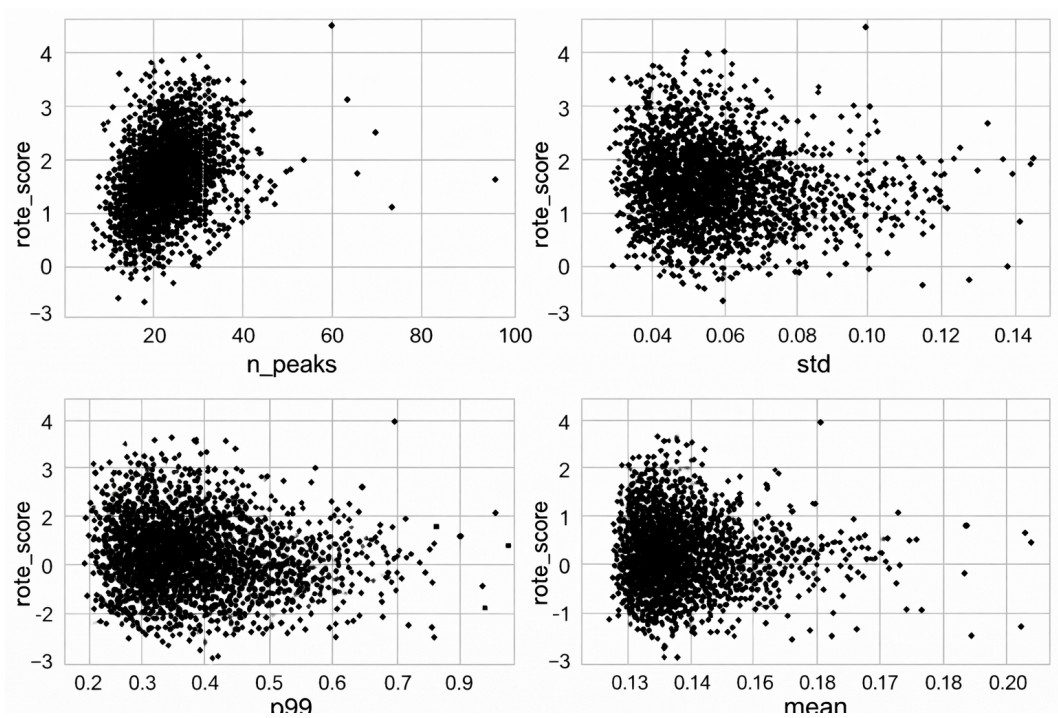

Figure 8: Scatter plots showing how the `rote_score` (activation along the top IOR/rote vector) relates to crude, interpretable image proxies: `n_peaks` (a crowding/source-count proxy), `std` (overall contrast/noise), `p99` (bright-tail intensity), and `mean`(average background level). The clearest trend is a positive relationship between `rote_score` and `n_peaks`, meaning higher rote activation is associated with more detected bright peaks / denser fields. In contrast, relationships with `mean`/`p99`/`std` are weaker, suggesting the rote direction is not simply brighter image or higher background, but is more consistent with crowding/field complexity as a shortcut feature.

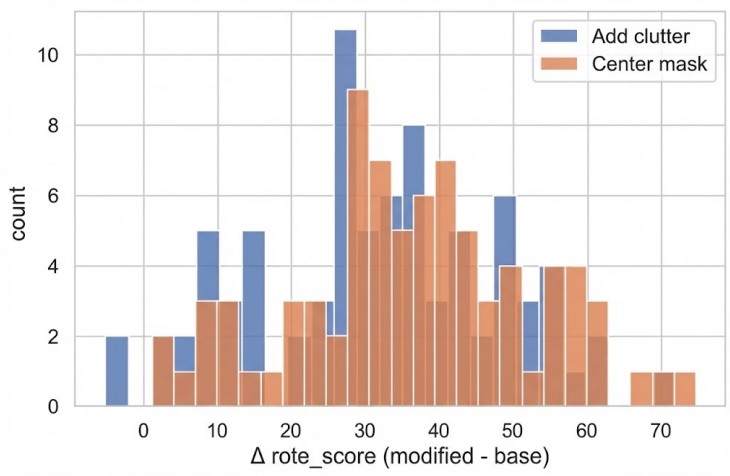

Figure 9: Intervention effect on rote_score. Both clutter injection and center masking shift the rote score upward (modified minus base).

## E.1    GEOMETRIC OVERLAP WITH PHOTOMETRY

We quantify overlap between the learned redshift direction $w_z$ and photometric directions.

Table 9: Directional overlap in DINOv2.

| Metric | Value | Meaning |
|---|---|---|
| $\cos(w_z, w_{\text{color}})$ | 0.542 | substantial overlap with color |
| $\cos(w_z, w_{\text{mag}})$ | 0.255 | moderate overlap with brightness |
| Containment in span(mag,color) | 33.4% | ~1/3 of $z$ signal lies inside photometry span |

### E.2 DINO z-SCORE TRACKS PHOTOMETRY AND SIZE

Figure 10 plots the DINO redshift probe score against real magnitude, real color, and a simple size proxy. The correlations are visible by eye: the redshift score is tightly coupled to photometric variables.

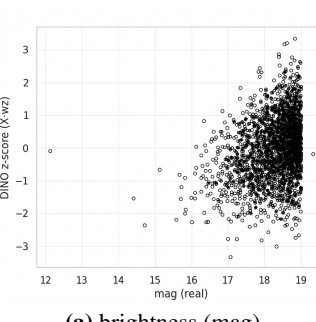 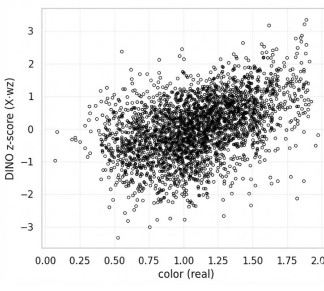 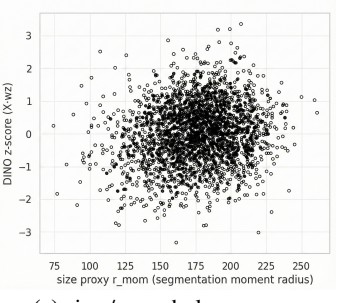

**(a)** brightness (mag)  **(b)** color  **(c)** size / morphology proxy

Figure 10: DINOv2: the learned redshift score co-varies with photometric variables and size/morphology

### E.3 RESIDUAL TEST: DOES DINO ADD REDSHIFT SIGNAL BEYOND PHOTOMETRY?

If DINO had a separate shortcut channel for redshift beyond photometry and simple morphology, then embeddings should explain a meaningful residual once we regress out photometry. We test this by comparing: (i) a predictor using only mag+color, (ii) a linear probe on DINO embeddings, and (iii) a residual probe that predicts what mag+color failed to explain.

Table 10: Residual test for DINOv2.

| Predictor | $R^2$ | Interpretation |
|---|---|---|
| Photometry only (mag + color) | 0.672 | direct measured correlates explain much of $z$ |
| DINO embeddings (linear) | 0.291 | compressed generic representation |
| Residual (embeddings beyond photometry) | 0.018 | near zero additional linear signal |

This supports the entanglement story: after accounting for photometry, DINO adds very little linearly accessible redshift information. In practice, this is why guarded erasure on DINO removes only a small fraction of $z$ while keeping photometry intact.

### E.4 DINO CONTAINS THE CLUTTER SIGNAL BUT IT IS NOT SEPARABLE

Even though DINO does not yield a clean removable rote direction under the photometry guard, it still encodes clutter/crowding statistics. We test this by predicting Qwen2-VL's `rote_score` from DINO embeddings and by correlating DINO's redshift score with Qwen's `rote_score`.

Table 11: DINOv2 contains Qwen's rote signal, but it is mixed into the same space as photometry.

| Metric | Value | Meaning |
|---|---|---|
| $R^2$ (predict Qwen `rote_score` from DINO emb) | 0.487 | clutter signal is linearly present in DINO |
| Pearson corr(DINO z-score, Qwen `rote_score`) | 0.654 | DINO redshift co-varies with clutter |
| Spearman corr(DINO z-score, Qwen `rote_score`) | 0.647 | monotonic association |

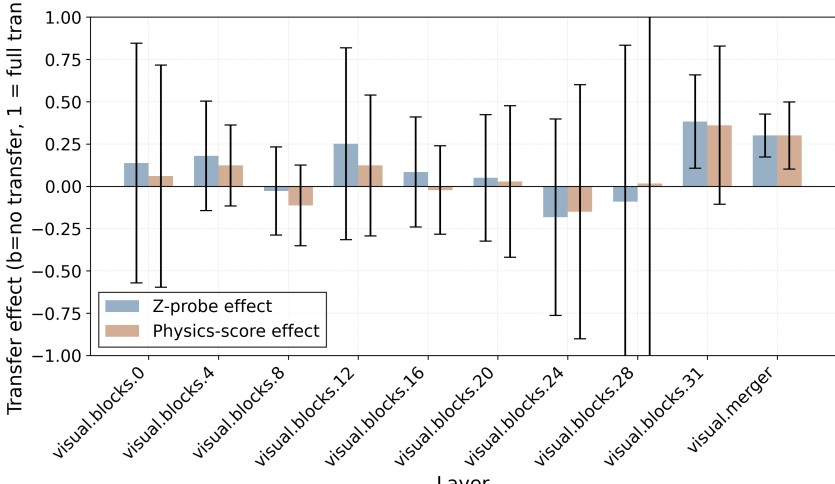

Figure 11: Causal tracing in Qwen2-VL (localized CLS patching, $\alpha = 0.3$). Transfer effects are small and unstable early, and become stronger in late blocks and the merger.

# F  CAUSAL PATHWAY MAPPING IN QWEN2-VL: PIXELS → FEATURES → LANGUAGE

The earlier sections establish that Qwen2-VL embeddings contain redshift information and that some shortcut-like component is partially removable. Here we ask where the usable distance signal forms inside the vision tower, and whether it causally affects downstream behavior.

## F.1  READOUTS

We track three readouts:

- **Z-probe readout:** a ridge probe from embeddings to $z$ (test $R^2 \approx 0.414$).
- **Physics score:** a 1D score $\langle h, v_{\text{physics}} \rangle$ along the learned distance direction.
- **Forced-choice language score:** $\log P(\text{"distant"}) - \log P(\text{"nearby"})$ under a fixed prompt format.

## F.2  LOCALIZED ACTIVATION PATCHING (CAUSAL TRACING)

We form near/far pairs from the test set (bottom/top $\sim 20\%$ in $z$). For a given vision layer, we cache the near-image activation (CLS-localized) and patch it into a forward pass of the far image using interpolation:

$$a_{\text{patched}} \leftarrow a_{\text{far}} + \alpha(a_{\text{near}} - a_{\text{far}}),$$

with $\alpha = 0.3$. We measure how much the far image's readout moves toward the near image. We report a normalized transfer effect where 0 means no transfer and 1 means full near→far transfer.

The pattern is simple: distance information is not cleanly usable from the start. Transfer becomes more consistent late in the vision stack and is consolidated at the merger.

Table 12: Causal tracing by layer (mean $\pm$ std across near/far pairs).

| Layer | Z-probe effect | Physics-score effect |
|---|---|---|
| visual.blocks.0 | $0.137 \pm 0.708$ | $0.060 \pm 0.656$ |
| visual.blocks.4 | $0.180 \pm 0.324$ | $0.123 \pm 0.239$ |
| visual.blocks.8 | $-0.028 \pm 0.261$ | $-0.113 \pm 0.239$ |
| visual.blocks.12 | $0.252 \pm 0.567$ | $0.123 \pm 0.416$ |
| visual.blocks.16 | $0.084 \pm 0.325$ | $-0.022 \pm 0.262$ |
| visual.blocks.20 | $0.050 \pm 0.374$ | $0.028 \pm 0.448$ |
| visual.blocks.24 | $-0.183 \pm 0.580$ | $-0.151 \pm 0.751$ |
| visual.blocks.28 | $-0.091 \pm 0.924$ | $0.017 \pm 1.028$ |
| visual.blocks.31 | $0.382 \pm 0.276$ | $0.361 \pm 0.468$ |
| visual.merger | $0.300 \pm 0.127$ | $0.300 \pm 0.198$ |

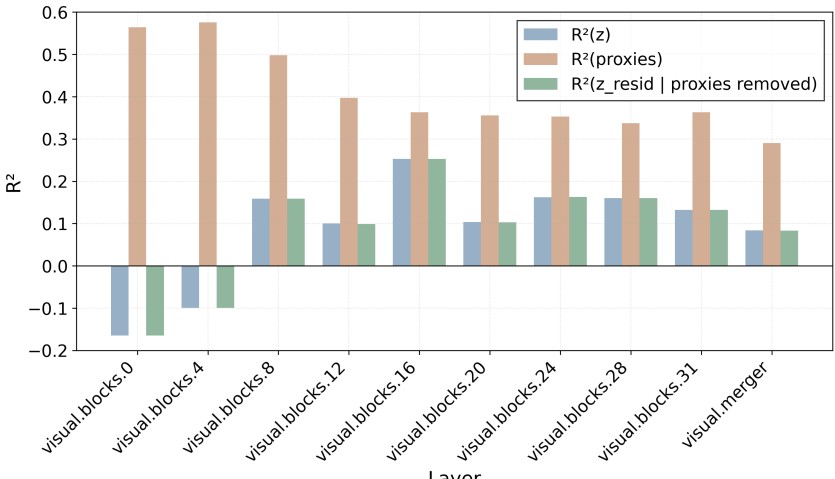

Figure 12: Causal scrubbing baseline (pixel proxies). Early layers strongly encode proxies; $z$ becomes decodable in mid/late layers.

### F.3 LANGUAGE FORCED-CHOICE UNDER PATCHING

We repeat the same patching experiment and measure the forced-choice language score. In this prompting format, the model has a strong baseline bias toward distant for both near and far images. Patching visual activations shifts the language score only slightly.

Table 13: Forced-choice language sensitivity under patching. Similar scores across conditions indicate weak coupling between patched visual distance signal and this forced-choice decision.

| Condition | Mean score | Std |
|---|---|---|
| Baseline (Far images) | 3.413 | 1.020 |
| Baseline (Near images) | 3.125 | 0.943 |
| Patched Far w/ Near at block 0 | 3.300 | 0.630 |
| Patched Far w/ Near at block 20 | 3.394 | 0.666 |
| Patched Far w/ Near at merger | 3.231 | 0.569 |

This creates a useful separation: yes, distance signal is causally present for embedding-level readouts at late layers; but that signal is not strongly coupled to the forced-choice language output under this prompt.

Table 14: Causal scrubbing baseline by layer. Early layers encode proxy statistics; $z$ becomes linearly decodable later.

| Layer | $R^2(z)$ | $R^2$(proxies) | $R^2(z_{\text{resid}})$ |
|---|---|---|---|
| visual.blocks.0 | -0.165 | 0.564 | -0.165 |
| visual.blocks.4 | -0.099 | 0.576 | -0.099 |
| visual.blocks.8 | 0.159 | 0.498 | 0.159 |
| visual.blocks.12 | 0.100 | 0.397 | 0.099 |
| visual.blocks.16 | 0.253 | 0.363 | 0.253 |
| visual.blocks.20 | 0.104 | 0.356 | 0.103 |
| visual.blocks.24 | 0.162 | 0.353 | 0.163 |
| visual.blocks.28 | 0.160 | 0.337 | 0.160 |
| visual.blocks.31 | 0.132 | 0.363 | 0.132 |
| visual.merger | 0.084 | 0.290 | 0.083 |

## F.4 CAUSAL SCRUBBING SUFFICIENCY TEST: $z$ BEYOND PIXEL PROXIES

Finally, we run a controlled "proxy baseline". We define simple pixel proxies (mean brightness, $p95$, and a crude size/area statistic) and measure: $R^2(z)$, $R^2$(proxies), and $R^2(z_{\text{resid}})$ where $z_{\text{resid}}$ is the residual after regressing $z$ on proxies.

This is a minimal baseline. It shows that early layers are dominated by simple pixel-level statistics, while $z$ becomes decodable later. A stricter sufficiency test can be to use calibrated astrophysical proxies (mag/color/size) as in Sections D.2 and E; we treat this proxy-scrubbing result mainly as a controlled pixel-level reference point.