# OpenReview forum: "Astronomy as a Ground-Truth Sandbox for Interpreting Large Models"
_ICLR.cc/2026/Workshop/Sci4DL — Submitted to Sci4DL 2026_

### Official Review · Reviewer_X1Ss · 2026-02-20

**Fit:** 2
**Significance:** 2
**Confidence:** 2

**Summary:**

This paper analyzes three different models, a vision only foundation model (DINOv2), a vision language model (Qwen2-VL), and an astronomy specific model (AstroPT), on the task of predicting distance of objects in astronomical images. This is posed as a way of studying mechanistic interpretability techniques in a setting where the target property is precisely quantifiable. The paper explores the dimension of subspaces that capture distance, whether these representations are transferrable, whether they can be used for predictable steering, and whether internal features decompose between those that are physically grounded and those that are spurious.

**Strengths:**

- Mechanistic interpretability techniques tend to be applied most often in either toy models (e.g., modular arithmetic) or generalist models (e.g., LLMs). It is interesting then to look at an extensive interpretation of a specialist model from science.
- The paper manages to cover a lot of ground in 4 pages. The reviewer would actually suggest using some of this space for more discussion of implications, but it cannot be argued that too little content is presented.

**Suggestions:**

- The astronomy connection is not sufficiently justified: The paper suggests using astronomy as a testbed for mechanistic interpretability. The reviewer agrees that exploring mechanistic interpretability within a verifiable domain is a nice idea. But the specific choice of astronomy itself is never justified. Past interpretability papers have used mathematics, basic physics, why not do the same here?
- The paper is missing the ‘why’ part. It makes lots of potentially interesting observations but never really addresses why the reader should care. In the abstract it claimed the paper introduces a ‘comprehensive testbed’. The reviewer feels that such a testbed would explore many different properties, not just distance. Thus, the reader goes in expecting a description of an expansive testbed. Instead they are given a very long analysis of three model’s representation of one property.
- “If a distance concept is really tied to the world, we might expect some overlap across these very different models”, I found this confusing. We know distance is part of the world. Also, one could imagine that there may be one type of model that has a high quality world model for physical properties in astronomy. Such a model might have a much better (and different) representation of distance, but this doesn’t say anything about whether a distance concept is “tied to the world”.

**Nitpicks:**
- Why is the Nanda citation in the abstract?
- Use of comma separation in large numbers should be consistent (e.g., line 050 and 052).
- Line 056: “the way we expect them to do.” -> “the way we expect them to.”
- Line 057: “…different in…” -> “…have different…”
- Line 084: The paper says there is a reason for each method being used but only one of these reasons is given.

---

### Official Review · Reviewer_W2CL · 2026-02-24

**Fit:** 3
**Significance:** 1
**Confidence:** 2

**Summary:**

- Authors propose an experimental framework/testbed for mechanistic interpretability based on an existing dataset of galaxy images and spectroscopic redshifts that measure astronomical distance.
- Below are the four main RQs and their answers, according to the authors.
	- RQ1: Does redshift behave like a single direction or a higher/lower-rank subspace?
		- A1: Redshift is linearly decodable but concentrated in a low-rank subspace.
	- RQ2: Does cross-model geometric alignment ==> useable feature transfer?
		- A2: No, cross-model geometric alignment does not necessarily imply linear features will transfer.
	- RQ3: Can diagnostic directions can be used for steering behaviour?
		- A3: Yes, can use diagnostic direction related to distance to steer language related to distance.
	- RQ4: Does removing direction or subspace constitute deletion of redshift concept?
		- A4: No, redshift is still recoverable after deleting direction of concept.
- Dataset has galaxy images and values for redshift. Redshift is a proxy to distance, which is the concept the authors are trying to understand.

**Strengths:**

- The lack of clear evaluation protocols for mechanistic interpretability work is a very important problem. The community's current focus on NLP means that, we are often dealing with experimental setups where the ground truth is ambiguous, as the authors point out. There is clearly a need for more rigorous experimental setups and natural science applications like astronomy seem like a reasonable place to start.
- The four RQs at the beginning and end of the paper help frame the results. The results themselves are interested but the claims seem overstated in some cases (see suggestions below).
- Overall I think the work is important and I would love to read an improved version of this paper where the experimental setup is more clearly articulated and the claims are more calibrated.

**Suggestions:**

- The authors assume way too much prior knowledge on both mechanistic interpretability and astrophysics. I understand that there are only four pages but then details should be put in the appendix instead of left out altogether. The paper as it currently stands is quite difficult to get through.
- The logic of the experimental setup needs to be better clarified. For example, take the results of the linear decodability experiment:
	- $\delta_{R^2) = R^2_{full} - R^2_{1D}$. Larger delta ==> larger difference in correlation with target between full vs 1D ==> 1D representation loses a lot of info compared to full representation==> concept is not linear. Is that the correct interpretation? If so, this should be spelled out for the reader.
- It is not clear to me whether the authors do any model training using this dataset or if the authors are using pretrained models. From reading the model descriptions on the second page, it looks like these are just pretrained models. If there is no training, then why are there two separate datasets (one with 74925 and another with 86500 examples)?
- There should be a clear description of what a PLS physics subspace is.
- There are some statements that are too bold e.g., "Our results show that physical concepts often reside in a higher-rank manifold than a single direction", even though the authors are only looking at one physical concept: redshift. Please be specific in your claims and avoid broad overstatements.

---

### Official Review · Reviewer_rGEi · 2026-02-27

**Fit:** 3
**Significance:** 2
**Confidence:** 1

**Summary:**

The paper argues that mechanistic interpretability often suffers from “fuzzy” targets (e.g., honesty/deception) with ambiguous ground truth, and proposes astronomical observations—specifically galaxy redshift as a physically defined proxy for distance—as a controlled sandbox to make interpretability claims more falsifiable.

**Strengths:**

A major strength of this paper is the framing: by using astronomical observations with physically defined, independently measured targets (galaxy redshift as a proxy for distance), it directly addresses the ambiguity of interpretability work on "fuzzy" concepts and makes claims more readily falsifiable under controlled manipulations.  Equally strong is the multi-model comparison across a vision-only foundation model (DINOv2), a vision-language model (Qwen2-VL), and an astronomy-specialist model (AstroPT), which reveals differences in how the same grounded quantity is represented.

**Suggestions:**

The paper would benefit from more details on the experimental setup: clearly document the data splitting protocol, the exact layers and pooling/extraction interfaces used per model, and all probe hyperparameters. In addition, many key claims appear to rest on single runs, so the authors should report variability over seeds with confidence intervals for main results.

---

### Meta-Review · Area_Chair_QuS1 · 2026-03-01

**Recommendation:** Reject

**Metareview:**

The paper makes broad claims without sufficient evidence and lacks details on the experimental setup which makes it difficult to understand the results and the scope of the work. I recommend a reject.

---

### Decision · Program_Chairs · 2026-03-02

Reject